# Relationship of taxonomic error to frequency of observation

**James B. Stribling** *, **Erik W. Leppo**

Tetra Tech, Incorporated Center for Ecological Sciences, Owings Mills, Maryland, United States of America

* james.stribling@tetratech.com

**Data Availability Statement:** All relevant data are within the paper and its Supporting Information files.

**Funding:** Approximately 10% of necessary level of effort in initiating this project was contracted to Tetra Tech, Inc. (JBS) (EP-C-14-016, Work

## Abstract

Biological nomenclature is the entry point to a wealth of information related to or associated with living entities. When applied accurately and consistently, communication between and among researchers and investigators is enhanced, leading to advancements in understanding and progress in research programs. Based on freshwater benthic macroinvertebrate taxonomic identifications, inter-laboratory comparisons of >900 samples taken from rivers, streams, and lakes across the U.S., including the Great Lakes, provided data on taxon-specific error rates. Using the error rates in combination with frequency of observation (FREQ; as a surrogate for rarity), six uncertainty/frequency classes (UFC) are proposed for approximately 1,000 taxa. The UFC, error rates, FREQ each are potentially useful for additional analyses related to interpreting biological assessment results and/or stressor response relationships, as weighting factors for various aspects of ecological condition or biodiversity analyses and helping set direction for taxonomic research and refining identification tools.

## Introduction

[1] discuss biodiversity in terms of not only richness of genotypes, species, and ecosystems, but also evenness of spatial and temporal distribution, functional characteristics, and their interactions. The sheer magnitude of biological species richness is largely unknown, with estimates ranging from 3–100 million [2–7]. For almost 300 years, efforts to organize and understand that diversity have used nomenclature and classification to provide a direct pathway to actual and conceptual catalogues of information about the biota; it is a system that can conceptually and functionally be thought of as a card catalogue in a library. With growing acceptance of the reality of global change and degradation in climate and both small- and large-scale habitat, along with diminishing taxonomic expertise, the task to census and record biota seems ever more daunting and urgent. Increases in computing power, information technology, and molecular techniques are encouraging some optimism in biodiversity research [8–13]. Even with some of these advances, progress in understanding biological diversity is uneven across taxonomic groups representing different segments of the tree of life, the bias mostly reflecting differential research attention and uneven sampling for some taxa in selected geographic areas [5, 14, 15].

Routine biological monitoring and assessment is about gathering representative sample data from defined habitat and using them for quantitative inference of environmental

Assignment 4-13) by the US Environmental Protection Agency/Office of Water/Office of Wetlands, Oceans, and Watersheds/Assessment and Watershed Protection Division. The work was in support of the Agency's National Aquatic Resources Surveys: https://www.epa.gov/national-aquatic-resource-surveys. Additionally, Tetra Tech, Inc., the employer of JBS and EWL, allowed some company resources to be applied to some of the data analyses and manuscript preparation, in particular, computer hardware, software, and network resources, and limited labor hours. The sponsors played no other role in the study design, data collection and analysis, decision to publish, or preparation of the manuscript.

**Competing interests:** Tetra Tech, Inc., the employer of JBS and EWL, allowed some company resources to be applied to some of the data analyses and manuscript preparation. This does not alter our adherence to PLOS ONE policies on sharing data and materials. No individuals, agencies, or private firms have interests in this work relating to employment, consultancy, patents, products in development, or marketed products.

conditions [16, 17]. Though such monitoring is not about documenting biodiversity or even absolute richness, the two fields rely on identical basic data as input for indicator calculations, model building, and decision-making, that is, taxonomic identifications. The name of an entity or object, whether individually or as a group or class, associates it with information on observable characteristics, provides answers to questions, and potentially allows new lines of enquiry to be framed and pursued. It is as much a truism of biological taxonomy as it is of basic human language that inconsistency in terminology impedes understanding and progress.

Historical development of biological nomenclature and classification has been considered by anthropologists as a fundamental component of language. Efforts to understand folk taxonomies have been through debating the relative merits of intellectualism vs. utilitarianism [18–20], approximating the difference between, respectively, basic curiosity and material need. The greater frequency with which an object is observed, there is improved reliability and consistency in its recognition, potentially leading to greater refinement of naming conventions/nomenclatural structure. In this context, it is important to define what is intended by labelling an object (or a taxon) as rare. From a theoretical perspective, rarity has been defined using niche- or phylogenetic-based concepts of abundance, distribution, rarity, or conservation priority-setting [21–24]. As an operational descriptor, rarity or relative commonness is frequency of encounter or observation.

The first principle and purpose of taxonomic identification and nomenclature is communication, and logically, objects that are more frequently observed will be recognized with increasing speed, reliability, and consistency. Biologist and ecologist perceptions of the relative rarity or commonness of taxa is a combination of life history and encounter frequency. As an example, reliability of botanical nomenclature used by the lay community in Chiapas, Mexico, was evaluated and use of plant names was found to be strongly related to cultural significance [25]. Techniques for communicating about plants with low cultural significance receiving little human attention were imprecise, that is, under-differentiating. Those with moderate cultural significance had a folk taxonomy which came closer to biological taxon definitions; and the extreme, plants with a high cultural significance tended to be over-differentiated. There is a conceptual relationship between cultural significance and familiarity, the latter of which would be enhanced by a high frequency of encounters/observation.

[26] developed a system for distribution classes of benthic macroinvertebrates, based on frequency of occurrence in the Netherlands. Using a combination of species rarity or commonness in their national dataset and direct input from a group of selected taxonomists, they developed a system comprising six different classes (Table 1). One of the driving factors behind their analysis was to have a classification system that would contribute to decision-making relative to conservation of aquatic resources.

**Table 1. Distribution classes describing relative rarity and commonness of benthic macroinvertebrates in the Netherlands [26].**

| Distribution class[1] | Percentage of sites |
|---|---|
| Very rare | 0–0.15 |
| Rare | 0.16–0.5 |
| Uncommon | 0.6–1.5 |
| Common | 1.6–4.0 |
| Very common | 4.1–12 |
| Abundant | >12 |

[1]Class definitions are based on frequency of occurrence, calculated as the percentage of sites.

Routine taxonomic quality control (QC) analysis used by the USEPA National Aquatic Resources Surveys (NARS) and several state, regional, and local monitoring programs for benthic macroinvertebrate samples are based on direct inter-laboratory comparisons. Randomly selected samples are identified by independent taxonomists, resulting in quantitative descriptors of data quality, error rates and potential causes, and information used for formulating corrective actions. A secondary use/added benefit of these analyses is that taxon-specific error rates are produced that can be used as direct indicators of taxon uncertainty, as weighting factors during calculation of quantitative indicators, to help guide development of tools for biological monitoring, in general, and taxonomic identification, in particular. The purpose of this paper is to present the process used for deriving the uncertainty values using morphology-based taxonomic identifications, discuss and summarize the results, and provide recommendations for their application and next step analyses.

## Methods

Data used in this analysis are from freshwater benthic macroinvertebrate samples, collected from rivers, streams, and lakes across the U.S., including the Great Lakes. All taxonomic identifications were executed in laboratories using necessary sample/specimen preparation techniques, optical equipment, and appropriate technical literature. The level of effort expended by taxonomists for identifications is standardized for individual programs or projects, and is typically genus level, with occasionally more coarse targets for selected taxa. The taxonomic comparison process used for routine QC analysis is described in detail elsewhere [27–29] and involves blind sample reidentification by independent taxonomists in separate laboratories of a randomly selected 10% of each sample lot.

We compiled interlaboratory comparison data for 914 samples from 10 large programs or projects (Table 2) which are conducted at selected local, regional, State, and National scales. Samples used by each of the programs for QC analyses [27, 30] were randomly selected from the full sample load of the program, typically at a rate of approximately 10%. Thus, results reported here can be considered as representative of more than 9,000 samples. There is a total

**Table 2. Datasets compiled and used in this analysis.**

| Entity | Project/Program Name | Sample years | No. samples[1] |
|---|---|---|---|
| Maryland Department of Natural Resources | Maryland Biological Stream Survey (MBSS) | 1995–2014 | 135 |
| Mississippi Department of Environmental Quality | Mississippi Phased Biological Monitoring | 2002–2018 | 133 |
| Prince George's County (MD) Department of the Environment | Watershed-Scale Biological Monitoring and Assessment Program | 2004–2017 | 54 |
| US Environmental Protection Agency/Office of Water (USEPA/OW) (National Survey) | Wadeable Streams Assessment (WSA_2004) | 2004 | 71 |
| US Army Corps of Engineers-Mobile District | Lake Allatoona/Upper Etowah River Watershed (LAUE) (GA, US) | 2007–2009 | 19 |
| USEPA/OW (National Survey) | National Lakes Assessment (NLA_2007) | 2007 | 96 |
| USEPA/OW (National Survey) | National Rivers and Streams Assessment (NRSA_2008) | 2008 | 134 |
| USEPA/OW (National Survey) | National Coastal Condition Assessment (NCCA_2015) (Great Lakes only) | 2015 | 49 |
| USEPA/OW (National Survey) | National Lakes Assessment (NLA_2017) | 2017 | 120 |
| USEPA/OW (National Survey) | National Rivers and Streams Assessment (NRSA_2018) | 2018 | 103 |
| **TOTAL** | | | **914** |

The number of samples generally represents approximately 10% of the entire sample load for each program during the indicated time period.

of 1,003 taxa, primarily at genus level (Fig 1), but also including more coarse levels because the level of effort was limited by defined standard procedures and/or poor specimen condition. Following Genus at 79.9 percent, the most frequently used categories were Family (14.6 percent), and Order and Subfamily (1.9 and 1.6 percent, respectively); other levels represent <1 percent of the dataset. There are occasionally "slash taxa", such as *Cricotopus/Orthocladius* (Diptera: Chironomidae), and one genus-group taxon, *Thienemannimyia* genus group which includes the chironomid genera *Conchapelopia*, *Rheopelopia*, *Helopelopia*, *Telopelopia*, *Meropelopia*, *Hayesomyia*, and *Thienemannimyia*. Truncatelloidea (Mollusca: Gastropoda) is used as a grouping for all Hydrobiidae. Two informal/undefined groupings were used: "Tubificoid Naididae" for those taxa formerly identified as Tubificidae (Oligochaeta: Haplotaxida); and Hydracarina for water mites that could not be taken to genus level.

Two different taxon-specific characteristics are quantified, frequency of observation, or relative rarity, and relative percent difference (RPD). The total number of individuals (count) for a given taxon is the sum across all primary taxonomists (T1), from all samples in all projects. That count is derived in the same manner for the QC taxonomists (T2). *Frequency of observation* ([FREQ] relative rarity, commonness) for a taxon is the percentage of samples for which a taxon was recorded, calculated as the number of samples in which the taxon was found relative to the total number of samples (n = 914). The number of samples for each taxon is the average between T1 and T2. We plotted numbers of taxa versus numbers of samples using logarithmic scales to illustrate the dominance of taxa observed in a single sample.

The proportional difference between two taxon-specific values is calculated using RPD [31] as an indication of the confidence with which a data user can rely on an identification result. It

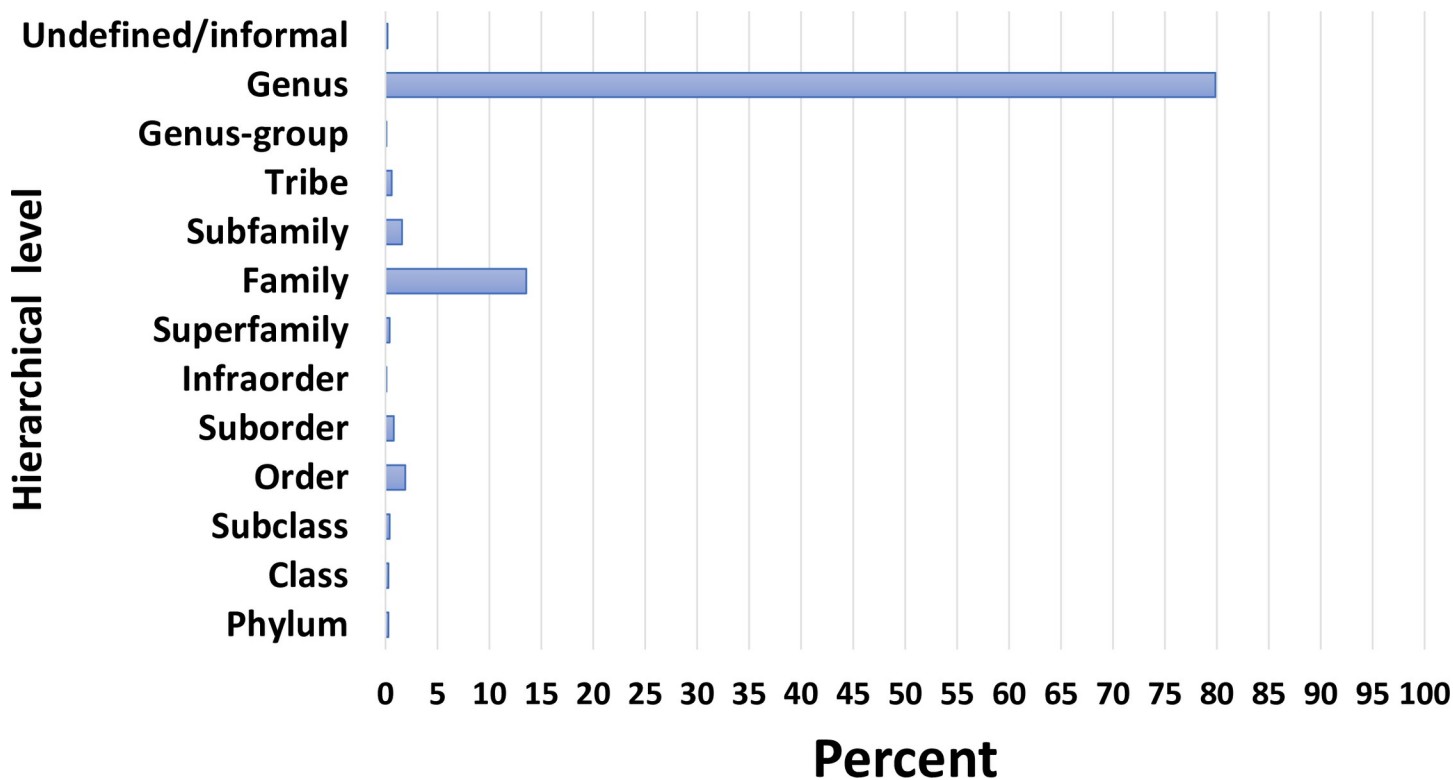

**Fig 1. Frequency distribution of taxa among hierarchical levels in this dataset.**

is calculated as follows:

$$RPD = \left( \frac{|A - B|}{(A + B)/2} \right) * 100$$

where *A* and *B* are the numbers of individuals counted for a taxon by T1 and T2, respectively, and pooled across all samples and projects. Values range from 0, indicating perfect agreement, to 200, or perfect disagreement. A general characteristic of RPD is that low values indicate better consistency of identifications between/among taxonomists, thus conveying greater certainty than high values.

Caution is warranted in using RPD when taxon-specific *counts* are low. If either T1 or T2 recorded ≥1 specimen of a taxon, and the other found none (0), RPD would be 200%. Although the number itself (200) would not be informative, it would indicate that one of the taxonomists recognized individuals of a taxon where the other did not. This would be a clue that some morphological key character (and, thus, the taxon) is not being recognized, or incorrect nomenclature is being applied. Other than these cautions, low values of RPD are reliable indicators of consistency. Thus, each taxon is represented by two data values, x = RPD and y = frequency of observation (FREQ) (S1 Appendix), as input for an x:y scatterplot. We used R-script to run a nonlinear regression model relating RPD to FREQ.

## Results

The first data visualization was to use a logarithmic plot of numbers of taxa versus numbers of samples (Fig 2). There are 304 taxa that are observed in only 1–2 samples, where the 33 most common taxa are found in anywhere from 200–674 samples. Seventy-five percent (75%) of the taxa were documented in ≤20 samples. Overall distribution ranged from 200 taxa each being found once (in a single sample), to one taxon, *Polypedilum* (Diptera: Chironomidae: Chironominae: Chironomini), occurring in 674 samples.

Taxon-specific RPD plotted against FREQ (Fig 3) illustrates that most taxa have low taxonomic uncertainty (mostly identified consistently) and are relatively infrequently encountered. The best fit nonlinear regression model is given by the exponential decay equation: RPD = 22.673 + (200.498)*e^(-0.192*FREQ), and all model terms were significant at p<0.001 (S1 Table). We delineated six uncertainty/frequency classes (UFC) based on graphic patterns (Figs 4 and 5), resulting in approximately 60% of taxa as being considered rare and identified with a high degree of certainty, that is, low RPD. All taxa are listed with associated numbers of individuals by primary and QC taxonomists, RPD, the number and percentages of samples, and UFC (S1 Appendix). Most taxa fall within UFC3 and 5 (Table 3; Figs 5 and 6), with roughly similar proportions within major taxa (Fig 7). UFC6 should be considered anomalous due to its representation by a small number of taxa (n = 6); otherwise, the mean and median values of RPD and FREQ, respectively, generally decrease and increase from UFC1-5 (Table 4, Fig 8).

We selected several taxa from each UFC (Table 5) to illustrate representative, quantitative outcomes and characteristics. UFC1 is **high confidence, common**, with representative taxa such as *Pisidium*, *Stenelmis*, *Caenis*, and *Hyalella*; overall, taxa in this class are observed in 23–74 percent of samples. Other than *Nais* with an RPD of 20.3, all other taxa in this class have RPD<10. UFC2 is **high confidence, moderately common**; overall, ranging in frequency of observation from 14–22 percent of samples, these taxa are also identified with low uncertainty (RPD, 0.3–17.6). Example taxa of this class include *Stempellinella*, *Baetis*, *Arrenurus*, and *Hemerodromia*. UFC3 groups taxa that are identified with confidence, simultaneous with being relatively rare (low frequency of occurrence) (**high confidence, rare**). Taxa range from

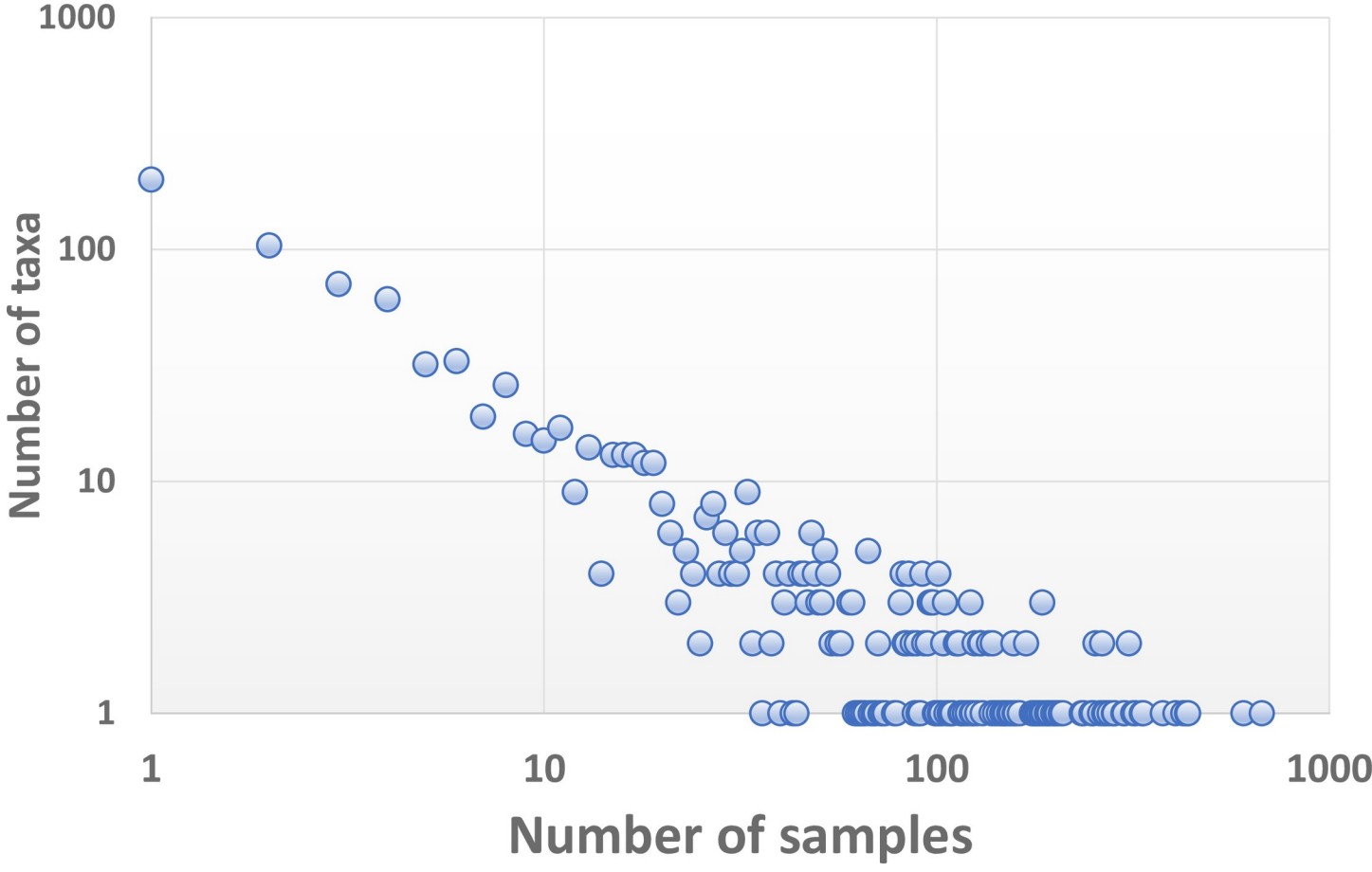

**Fig 2. Logarithmic scatterplot illustrating that most taxa in this dataset are infrequently observed.**

being observed in only a single sample (0.1 percent of total *n*), such as *Anchycteis*, *Susperatus*, *Marilia*, and *Armiger*, to just under 14 percent, 120–125 samples (*Stenonema*, *Chimarra*, *Limnesia*, *Stictochironomus*). UFC4 groups taxa that are identified with increased uncertainty and are uncommon (Fig 4) (**moderate confidence, rare**). RPD ranges from 55–82, and taxa represent 0.1 percent of the samples (*n* = 1) to 5.7 percent (*n* = 52). Examples of UFC4 taxa include *Halesochila*, *Vacupernius*, and *Macrelmis* from only a single sample to *Cernotina*, *Teloganopsis*, and *Micromenetus* (*n* = 11, 15, and 52 samples, respectively). UFC5 groups taxa that are simultaneously rare and identified with a high degree of uncertainty (**low confidence, rare**), with taxa being observed in from 0.1–4.3 percent of samples, and identification uncertainty ranging from 85.7–200 (S1 Appendix). Example UFC5 taxa of lowest observation frequency include *Amphicosmoecus* and *Kogotus* (*n* = 1 sample) to *Placobdella* and *Sphaerium* in 16 (1.8 percent) and 39 (4.3 percent) samples, respectively. UFC6 taxa are **outliers, mixed**, not clearly falling in the other classes; there are six in this dataset, three of which are genus level (*Conchapelopia*, *Thienemannimyia*, and *Dero*), and three, family (Polycentropodidae, Libellulidae, and Naididae).

Major taxa are most heavily represented in UFC3 and 5 (Table 6, Fig 6). Chironomidae (*n* = 104), Trichoptera (*n* = 72), Coleoptera (*n* = 68), Ephemeroptera (*n* = 59), and Plecoptera (*n* = 47), in descending order, are the top five major taxa in UFC3, while Coleoptera (*n* = 23),

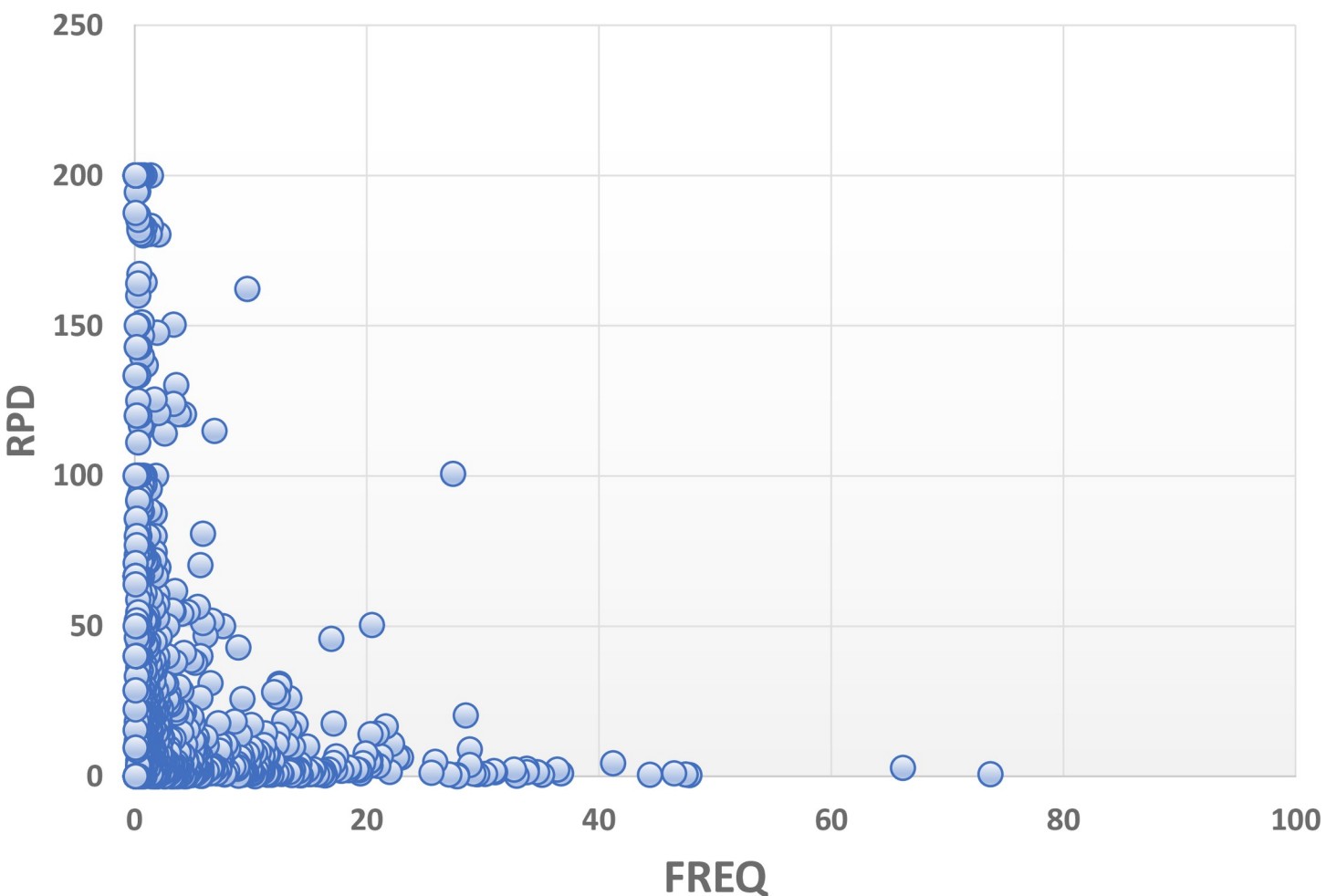

**Fig 3. Taxon-specific relative percent difference (RPD) plotted against frequency of observation (FREQ), or percent of total number of samples.**

Chironomidae (n = 21), Annelida (n = 20), Arachnida (n = 19), and Ephemeroptera and Plecoptera (tied, each n = 16) are those for UFC5.

## Discussion

Taxa with the highest RPD values, that is, with greater uncertainty, are documented in smaller numbers of sites (Table 7), corresponding with very rare and rare distribution classes of [23], and clearly illustrated by UFC1-2 versus UFC4-5 (Fig 7). In general, the more rare a taxon is, the greater is the uncertainty associated with its identity; and the obverse, increasingly common taxa are better known and identified with elevated confidence. This observation is demonstrated by the near mirror images of error rate (RPD) and rarity (FREQ) for UFC1-5 (Fig 8) and reflects the outcome predicted by [25], i.e., familiarity is borne of repeated encounters. This also speaks, in part, to the collective sense of our limited understanding of biological diversity, and of the most appropriate and effective ways of communicating about that diversity.

Higher level macroinvertebrate taxa in this analysis shown to have *greater identification confidence and consistency* are midges (Insecta: Diptera: Chironomidae), caddisflies (Insecta: Trichoptera), beetles (Insecta: Coleoptera), snails (Mollusca: Gastropoda), and stoneflies

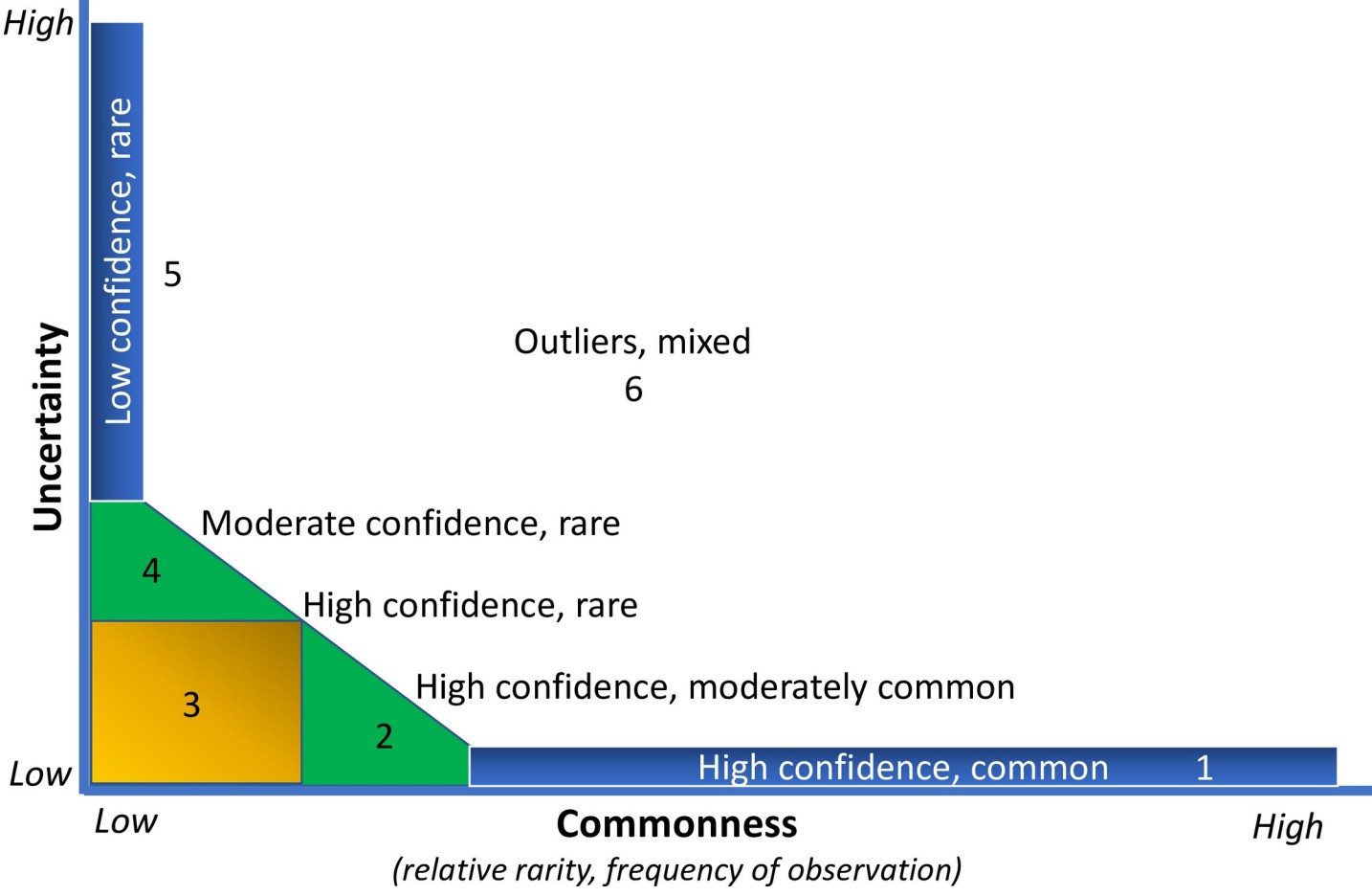

**Fig 4. Uncertainty/frequency model categories delineated relative to the graphic pattern shown in Fig 3.**

(Insecta: Plecoptera) (Fig 7), as they are mostly made up of finer level taxa within UFC1-3. Conversely, higher level taxa for which identification data seem to be *more problematic* (i.e., greater uncertainty) are bivalves (Mollusca: Bivalvia) and Crustacea (Arthropoda); these groups have a higher percentage of taxa in UFC4-5.

Several potential uses of UFC designations are relevant to informing data analysts and data users on the extent to which confidence can be placed in results. They include being used as taxon-specific weighting factors for calculating biological indicator values, such as indexes of biological integrity (IBI), River Invertebrate Prediction and Classification System (RIVPACS) models, various diversity calculations, species protection, or habitat prioritization. Testing is necessary to determine the effect on indicator values, but a weighted-average index could be formulated to elevate or restrict the importance of a taxon due to the relative potential of identification error. Similar to use of stressor tolerance values in the Hilsenhoff Biotic Index (HBI), UFC numbers could be used as taxon count modifiers. This approach would retain the inherent value and information content of organism identity, and simultaneously help objectively moderate the influence of those taxa on quantitative indicator outcomes.

Taxa demonstrated as having elevated identification uncertainty could be targeted for basic focused research, including morphological re-description, dichotomous identification keys, genetic fingerprinting, or other tools. Commonness values (FREQ) for individual taxa would

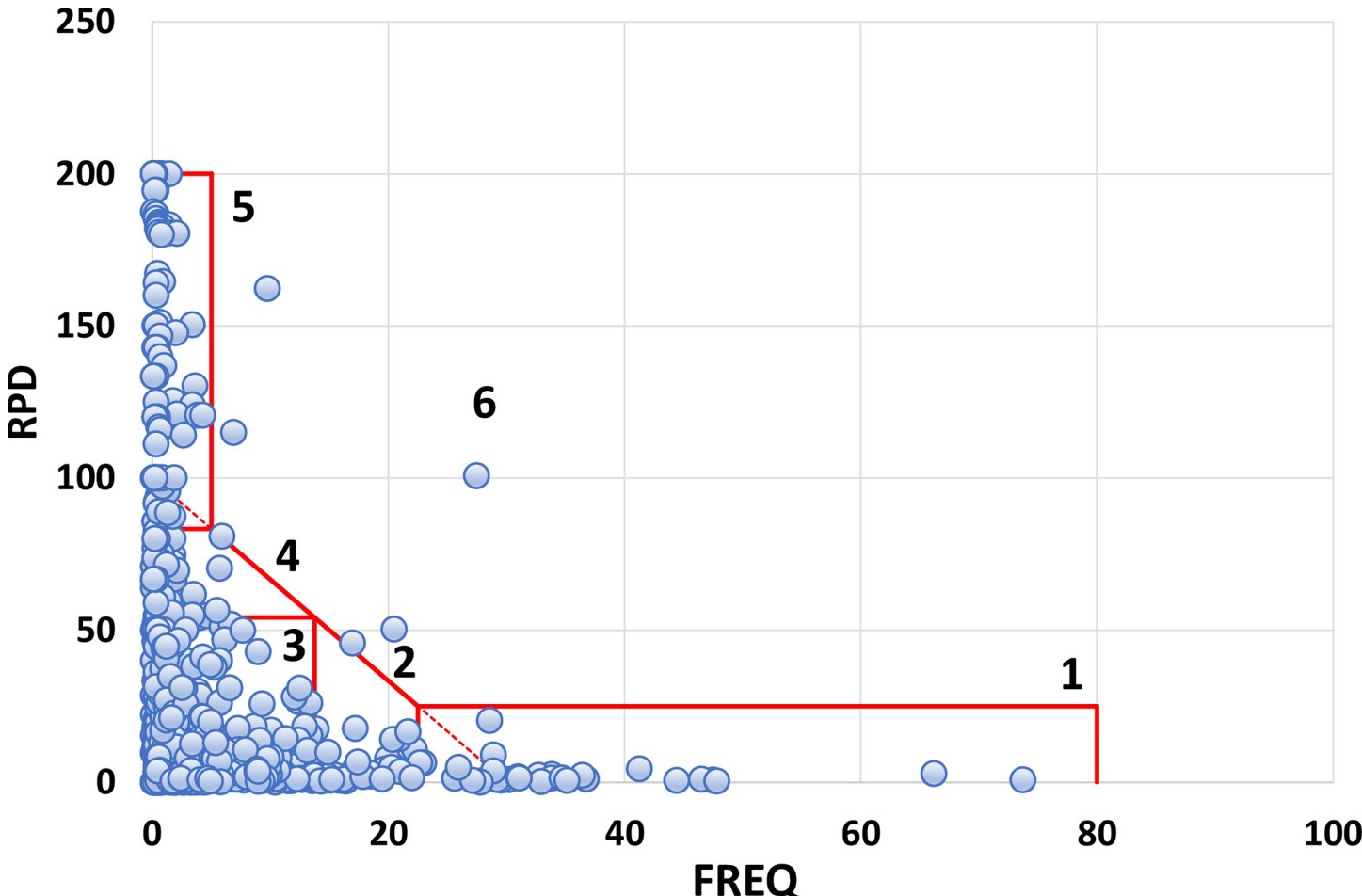

**Fig 5. Distribution of taxa within uncertainty/frequency categories (UFC1-6).** Uncertainty is expressed as relative percent difference (RPD) and relative rarity or commonness as frequency of observation (FREQ). Each point represents a taxon.

allow users of comprehensive identification manuals (such as, for example, [32]) to evaluate the relative rarity. The need for independent verification of an identification result would be emphasized for those with known elevated error rates (high RPD).

**Table 3. Identification uncertainty/frequency classes (UFC).**

| UFC | No. taxa (*n*) | Percent[1] | Taxonomic certainty | Commonness |
|---|---|---|---|---|
| 1 | 30 | 3.0 | High | Common |
| 2 | 40 | 4.0 | High | Moderate |
| 3 | 606 | 60.4 | High | Rare |
| 4 | 79 | 7.9 | Moderate | Rare |
| 5 | 242 | 24.1 | Low | Rare |
| 6 | 6 | 0.6 | Mixed | Mixed |

The confidence (certainty) placed in taxonomic identifications is related to both frequency of observation (commonness) and the consistency of identification.

[1]Percent is the percentage of taxa relative to the overall dataset.

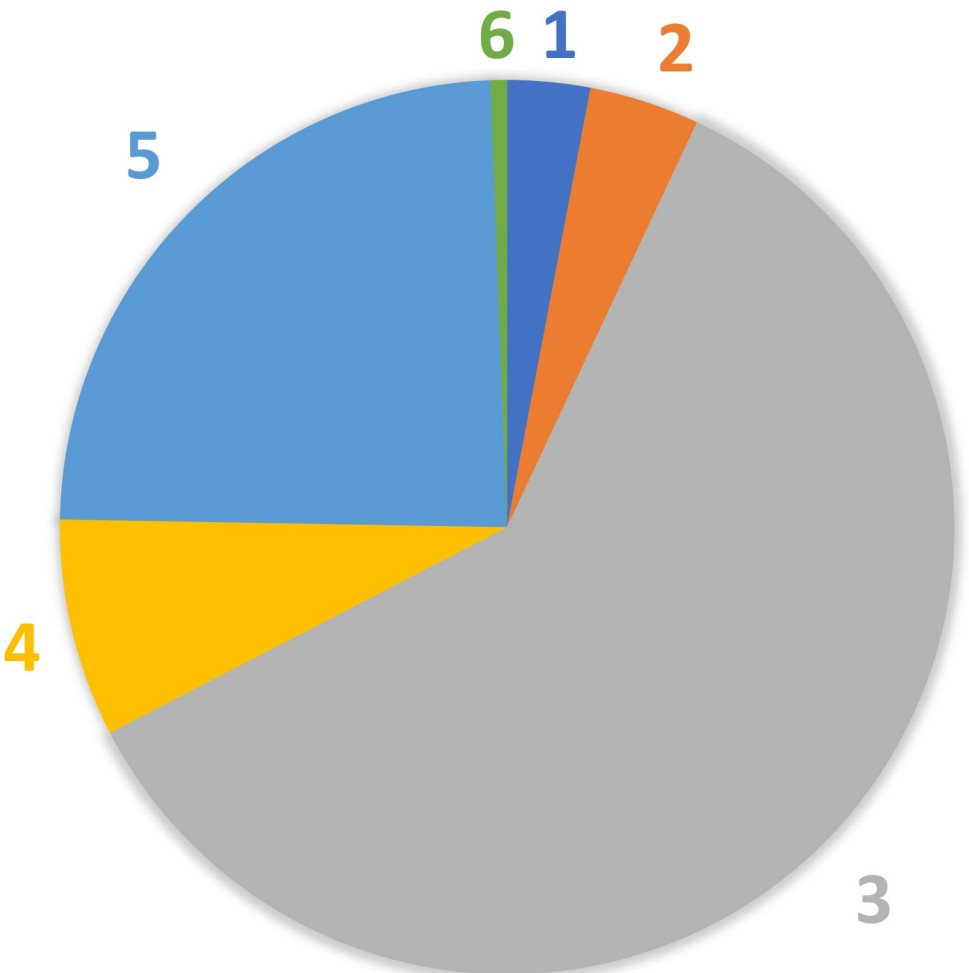

**Fig 6. Proportion of taxa falling within six uncertainty/frequency classes (UFC).** Approximately 67% of taxa are reliably identified with a high level of certainty (UFC 1–3), and 24.1% (UFC 5) are identified with a low level of certainty. Taxa within UFC 3 and 5 are also considered as rare or having a low frequency of observation.

Another potential use of these results would be in helping target individual taxa for determining causes, beyond lack of familiarity, of higher error rates. A common cause is known to be specimens in poor condition and/or small body size (early life stages, or instars). An outcome of such an investigation might be to specify standard procedures for some taxa, including for sampling, handling, preservation, and identification. An example of this would be a requirement that all larval Chironomidae be slide-mounted for examination under a compound microscope. We do not necessarily advocate this, as slide-mounting is not consistently needed by all laboratories or taxonomists. Rather, we stress that the taxonomist use whatever method is needed to attain the target taxonomic level as defined by program or study goals. The goal in this case is not to require that all taxonomists (or taxonomic technicians) slide-mount all chironomid midges; rather, the goal is to acquire genus level data for the taxon. In some cases, slide-mounting might be needed, in others, it would not. Thus, the need for such actions would be determined on a case-by-case, taxon-by-taxon, or even taxonomist-by-taxonomist basis, but the goal of genus level data remains the same.

## MAJOR TAXA

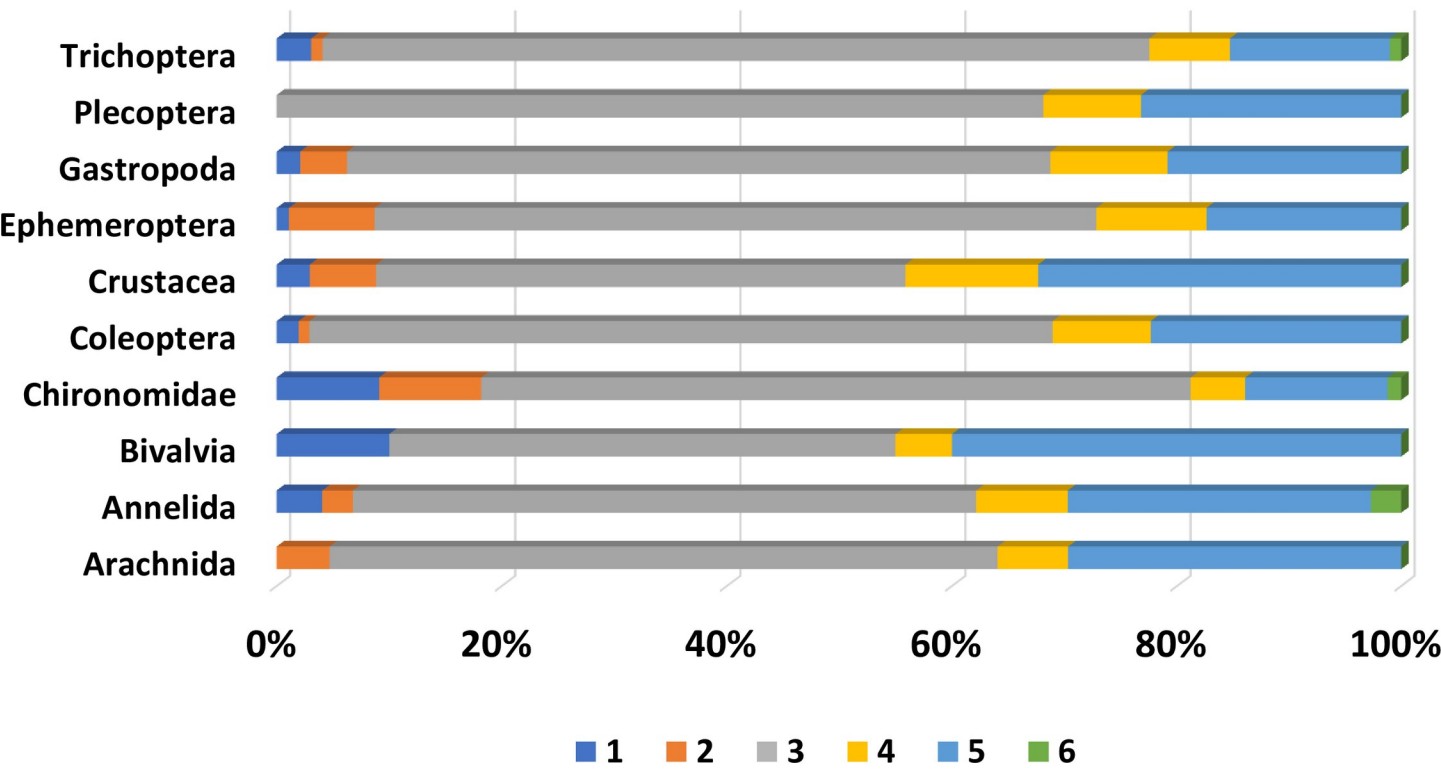

**Fig 7. Percentages of taxa in "major" benthic macroinvertebrate groups in six uncertainty/frequency classes.** 1, high confidence, common; 2, high confidence, moderately common; 3, high confidence, rare; 4, moderate confidence, rare; 5, low confidence, rare; 6, outliers, mixed.

**Table 4. Descriptive statistics for relative percent difference (RPD) and frequency of occurrence (FREQ).**

| UFC | RPD | | | | |
|---|---|---|---|---|---|
| | *Median* | *Mean* | *SD*[1] | *Min*[1] | *Max*[1] |
| 1 | 1.3 | 2.7 | 4.0 | 0 | 20.3 |
| 2 | 2.3 | 4.6 | 5.0 | 0.3 | 17.6 |
| 3 | 7.2 | 13.0 | 14.9 | 0 | 54.1 |
| 4 | 66.7 | 67.2 | 6.6 | 54.5 | 82.4 |
| 5 | 200 | 176.4 | 38.5 | 85.7 | 200 |
| 6 | 90.7 | 92.4 | 43.7 | 45.7 | 162.2 |
| | FREQ | | | | |
| 1 | 31.9 | 35.4 | 11.6 | 22.6 | 73.7 |
| 2 | 17.2 | 17.6 | 2.6 | 13.9 | 22.2 |
| 3 | 1.2 | 2.9 | 3.5 | 0.1 | 13.7 |
| 4 | 0.3 | 0.9 | 1.2 | 0.1 | 5.7 |
| 5 | 0.1 | 0.4 | 0.6 | 0.1 | 4.3 |
| 6 | 13.3 | 14.6 | 8.5 | 5.9 | 27.5 |

[1]SD is standard deviation, Min and Max are minimum and maximum.

Numbers of taxa (*n*) representing each class are given in Table 3.

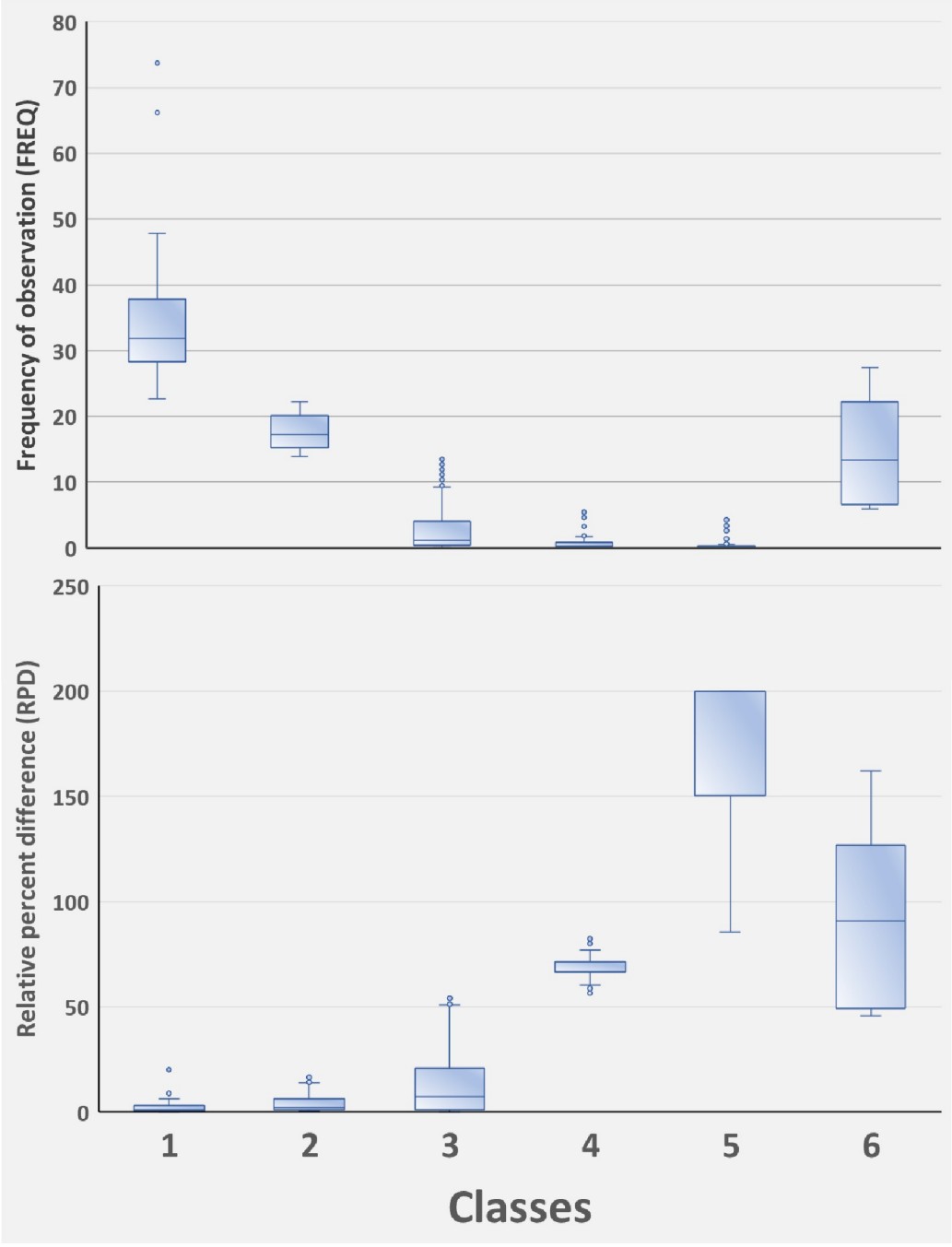

**Fig 8. Percentile distributions (boxplots) for frequency of taxon occurrence (FREQ) and relative percent difference (RPD) among the uncertainty-frequency classes.** FREQ is the percentage of samples for which a taxon was observed; RPD is a measure of uncertainty associated with taxonomic identifications, thus lower values equate to increased confidence. 1, high confidence, common; 2, high confidence, moderately common; 3, high confidence, rare; 4, moderate confidence, rare; 5, low confidence, rare; 6, outliers, mixed.

Our interest is in seeing UFC values used as one tool to enhance biological assessments, whether as direct input to indicator calculations, as information to help formulate additional analytical questions, or to help set or justify interpretive procedures. This analysis was possible

**Table 5. Selected taxa as representative examples of uncertainty/frequency classes (UFC).**

| UFC | Class | Order | Family | Genus | T1 | T2 | RPD | n | Pct. |
|-----|-------|-------|--------|-------|----|----|-----|---|------|
| 1 | Bivalvia | Veneroida | Pisidiidae | *Pisidium* | 1772 | 1888 | 6.3 | 207 | 22.6 |
| 1 | Insecta | Coleoptera | Elmidae | *Stenelmis* | 2507 | 2522 | 0.6 | 248 | 27.1 |
| 1 | Insecta | Ephemeroptera | Caenidae | *Caenis* | 16630 | 16562 | 0.4 | 437 | 47.8 |
| 1 | Malacostraca | Amphipoda | Hyalellidae | *Hyalella* | 15844 | 15875 | 0.2 | 301 | 32.9 |
| 2 | Arachnida | Trombidiformes | Arrenuridae | *Arrenurus* | 1869 | 1832 | 2.0 | 169 | 18.5 |
| 2 | Insecta | Diptera | Chironomidae | *Micropsectra* | 2172 | 2211 | 1.8 | 149 | 16.3 |
| 2 | Insecta | Ephemeroptera | Baetidae | *Baetis* | 4221 | 3999 | 5.4 | 186 | 20.4 |
| 2 | Insecta | Odonata | Coenagrionidae | *Enallagma* | 362 | 431 | 17.4 | 127 | 13.9 |
| 3 | Gastropoda | Basommatophora | Planorbidae | *Gyraulus* | 1992 | 1805 | 9.8 | 125 | 13.7 |
| 3 | Insecta | Diptera | Chironomidae | *Stictochironomus* | 1438 | 1406 | 2.3 | 125 | 13.7 |
| 3 | Insecta | Diptera | Simuliidae | *Prosimulium* | 1363 | 1363 | 0.0 | 95 | 10.4 |
| 3 | Insecta | Ephemeroptera | Leptohyphidae | *Tricorythodes* | 3961 | 4001 | 1.0 | 112 | 12.3 |
| 4 | Clitellata | Haplotaxida | Naididae | *Ripistes* | 11 | 6 | 58.8 | 6 | 0.7 |
| 4 | Crustacea | Isopoda | Asellidae | *Asellus* | 37 | 19 | 64.3 | 4 | 0.4 |
| 4 | Gastropoda | Basommatophora | Planorbidae | *Micromenetus* | 592 | 284 | 70.3 | 52 | 5.7 |
| 4 | Insecta | Plecoptera | Taeniopterygidae | *Oemopteryx* | 24 | 11 | 74.3 | 6 | 0.7 |
| 5 | Annelida | Lumbriculida | Lumbriculidae | *Stylodrilus* | 14 | 79 | 139.8 | 6 | 0.7 |
| 5 | Bivalvia | Veneroida | Sphaeriidae | *Sphaerium* | 460 | 114 | 120.6 | 39 | 4.3 |
| 5 | Insecta | Diptera | Ceratopogonidae | *Mallochohelea* | 14 | 93 | 147.7 | 18 | 2.0 |
| 5 | Insecta | Ephemeroptera | Siphlonuridae | *Siphlonurus* | 9 | 26 | 97.1 | 8 | 0.9 |
| 6 | Clitellata | Haplotaxida | Naididae | *Dero* | 2132 | 3565 | 50.3 | 187 | 20.5 |
| 6 | Insecta | Diptera | Chironomidae | *Conchapelopia* | 120 | 1149 | 162.2 | 89 | 9.7 |
| 6 | Insecta | Odonata | Libellulidae | | 237 | 64 | 115.0 | 63 | 6.9 |
| 6 | Insecta | Trichoptera | Polycentropodidae | | 179 | 76 | 80.8 | 54 | 5.9 |

The full list of 1,003 taxa is presented in S1 Appendix. T1 and T2 are the summed counts across *n* samples. RPD is relative percent difference, and Pct. is the percentage of total samples (*n* = 914) used in this analysis.

by having access to available output of inter-taxonomist comparisons and demonstrates added benefits of routine QC and operational data management routines.

**Table 6. Numbers of taxa by uncertainty/frequency class (UFC).**

| "Major" taxon | UFC (no. taxa) | | | | | | TOTAL |
|---------------|---|---|---|---|---|---|-------|
| | 1 | 2 | 3 | 4 | 5 | 6 | |
| Arachnida | 0 | 3 | 38 | 4 | 19 | 0 | 64 |
| Annelida | 3 | 2 | 41 | 6 | 20 | 2 | 74 |
| Bivalvia | 2 | 0 | 9 | 1 | 8 | 0 | 20 |
| Chironomidae | 15 | 15 | 104 | 8 | 21 | 2 | 165 |
| Coleoptera | 2 | 1 | 68 | 9 | 23 | 0 | 103 |
| Crustacea | 1 | 2 | 16 | 4 | 11 | 0 | 34 |
| Ephemeroptera | 1 | 7 | 59 | 9 | 16 | 0 | 92 |
| Gastropoda | 1 | 2 | 30 | 5 | 10 | 0 | 48 |
| Plecoptera | 0 | 0 | 47 | 6 | 16 | 0 | 69 |
| Trichoptera | 3 | 1 | 72 | 7 | 14 | 1 | 98 |
| *Total no. taxa* | *28* | *33* | *484* | *59* | *158* | *5* | *767* |

**Table 7. Relating relative percent difference (RPD) to distribution classes.**

| Nijboer and Verdonschot (2004) | | RPD (this study) | | | | | |
|---|---|---|---|---|---|---|---|
| Distribution class | Pct. of sites | n | Median | Mean | SD | Min | Max |
| Very rare | <0.16 | 200 | 200.0 | 139.2 | 88.0 | 0 | 200 |
| Rare | 0.16–0.5 | 235 | 40.0 | 61.9 | 63.8 | 0 | 200 |
| Uncommon | 0.6–1.5 | 181 | 21.6 | 44.8 | 53.8 | 0 | 200 |
| Common | 1.6–4.0 | 151 | 8.5 | 24.1 | 34.9 | 0 | 180.3 |
| Very common | 4.1–12 | 145 | 4.9 | 13.2 | 23.3 | 0 | 162.2 |
| Abundant | >12 | 91 | 2.3 | 7.6 | 13.8 | 0 | 100.7 |

"*n*" is the number of taxa that would be categorized as belonging to the [26] distribution classes based on frequency of occurrence in this study.

## Supporting information

**S1 Appendix. Uncertainty/frequency dataset, with benthic macroinvertebrate phylogenetic/classification hierarchy.** Primary and quality control counts (T1 and T2, respectively) are cumulative across n samples, relative percent difference (RPD), percent of samples, uncertainty/frequency class (UFC), and taxonomic rank.
(XLSX)

**S1 Table. Nonlinear regression of FREQ against RPD.**
(XLSX)

## Acknowledgments

Thoughtful reviews by colleagues provided conceptual and practical input and suggestions that helped improve the manuscript. We are especially grateful to Piet Verdonschot, Richard Mitchell, Ben Jessup, Chris Ruck, Mike Cole, Bern Sweeney, and John Morse. We also thank colleagues and clients from the USEPA Office of Wetlands, Oceans, and Watersheds; the Mississippi Department of Environmental Quality; Maryland Department of Natural Resources; Prince George's County (MD) Department of the Environment; and the US Army Corps of Engineers-Mobile District (Lake Allatoona/Upper Etowah River Watershed Partnership, Canton, GA). Subcontract laboratories involved with most of these comparisons include Freshwater Benthic Services (Petoskey, MI), Aquatic Resources Center (Nashville, TN), EcoAnalysts, Inc. (Moscow, ID), and Cole Ecological, Inc. (Greenfield, MA); at the time of sample identifications, all taxonomists were current with certifications from the Society for Freshwater Science's Taxonomic Certification Program. This manuscript was improved by comments from one anonymous reviewer.

## Author Contributions

**Conceptualization:** James B. Stribling.

**Data curation:** Erik W. Leppo.

**Formal analysis:** James B. Stribling.

**Funding acquisition:** James B. Stribling.

**Investigation:** James B. Stribling.

**Methodology:** James B. Stribling, Erik W. Leppo.

**Project administration:** James B. Stribling.

**Resources:** James B. Stribling.

**Software:** Erik W. Leppo.

**Supervision:** James B. Stribling.

**Validation:** James B. Stribling.

**Visualization:** James B. Stribling.

**Writing – original draft:** James B. Stribling.

**Writing – review & editing:** James B. Stribling.

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
