## [Decision Letter · Decision Letter 0]

8 Oct 2020

PONE-D-20-27394

Relationship of taxonomic error to frequency of observation

PLOS ONE

Dear Dr. Stribling,

Thank you for submitting your manuscript to PLOS ONE. After careful consideration, we feel that it has merit but does not fully meet PLOS ONE’s publication criteria as it currently stands. Therefore, we invite you to submit a revised version of the manuscript that addresses the points raised during the review process.

Editor's comments:  The reviewer was very enthusiastic about the manuscript but did have a number of minor suggestions about improvements that could be made prior to publication.  Please make as many of these as you can.

We look forward to receiving your revised manuscript.

Kind regards,

Judi Hewitt

Academic Editor

PLOS ONE

Journal Requirements:

"Approximately 10% of necessary level of effort in initiating this project was contracted to Tetra Tech, Inc. (JBS) (EP-C-14-016, Work Assignment 4-13) by the US Environmental Protection Agency/Office of Water/Office of Wetlands, Oceans, and Watersheds/Assessment and Watershed Protection Division. The work was in support of the Agency's National Aquatic Resources Surveys:  

https://www.epa.gov/national-aquatic-resource-surveys

The sponsors played no role in the study design, data collection and analysis, decision to publish, or preparation of the manuscript."

We note that you received funding from a commercial source: Tetra Tech, Inc

3. Please amend the manuscript submission data (via Edit Submission) to include author: Erik W. Leppo

Reviewers' comments:

Reviewer's Responses to Questions

**Comments to the Author**

1. Is the manuscript technically sound, and do the data support the conclusions?

Reviewer #1: Yes

2. Has the statistical analysis been performed appropriately and rigorously? 

Reviewer #1: Yes

3. Have the authors made all data underlying the findings in their manuscript fully available?

Reviewer #1: Yes

4. Is the manuscript presented in an intelligible fashion and written in standard English?

Reviewer #1: Yes

5. Review Comments to the Author

Reviewer #1: Manuscript Number PONE-D-20-27394

Relationship of taxonomic error to frequency of observation

Ecological monitoring and taxonomic assessment is fundamental for effective management of freshwater ecosystems. Error is rarely measured in biomonitoring surveys because it is often assumed to be small and constant, but is considerably more widespread than assumed. This short manuscript provides an interesting method of quantifying taxon-specific error rates for macroinvertebrate identification.

Macroinvertebrate identification requires a high level of skill and training, and misidentification of taxa, sorting efficiency and taxon enumeration can all impact data quality. Interpreting rare taxa data can be problematic for freshwater ecologists and, in my experience, assessing rare and often misidentified taxon are not handled consistently between regional water authorities and agencies.

Providing tools to better measure and quantify biodiversity values from national datasets, researchers can more accurately assess distinctiveness of rare taxon groups. This manuscript provides a means to better understand freshwater macroinvertebrate assemblages and, more importantly, highlight focus taxa groups that may require further investigation, better taxonomic keys, management or protection. Furthermore, this manuscript indirectly demonstrates the importance of implementing regular interlaboratory quality control procedures to identify causes and sources of error for macroinvertebrate samples. This will continue to ensure, sorting laboratories are able to produce consistent, high quality data.

I have some minor points to note, mostly with respect to the inconsistencies between formatting and presentation of the figures. However, some of these format issues may be a reflection of the journal requirements.

1. Lines 221, 226, 229, 230, 233-4, 239, 242 can you provide context to the taxonomic groups (e.g., order) the mentioned taxa belong, as done for Nais (line 222)? For a non-taxonomist or ecologist without knowledge of North America macroinvertebrates this would help immensely rather than the having to cross reference with supplementary files.

2. Although there is no mention to macroinvertebrate size (in relation to instar size) can you comment if instar size correlates to an increase in RPD error?

3. Lines 222, 224, 232. Consistency required for expressing percentages.

4. Figure 1. Can this axis extend to 100% as seen in Figure 5? Do this also for Fig. 3. The titles of X and Y axes are in upper case and not consistent with the other figures.

5. Figures 3, 4 and 5. There is considerably overlap in these three figures. Figure 3 is not necessary as this information is repeated in Figure 5. Although there is subtle difference in how these data points are plotted, they are essentially the same graph. Instead, can figures 4 and 5 be combined? For example, can each of the data points that correspond to a UFC categories be colour coded or superimposed on Figure 4?

6. Figure 6. A 2D rather than a 3D plot would be preferred. A legend is required (1 = High confidence, common; 2 = High confidence, moderately common etc) to avoid the reader having to reference back to Figure 4.

7. Figure 7. This figure needs to be produced in a similar format to Figure 1, i.e., consistent formatting of Y and X axis titles. As noted above, it is suggested a legend is added to this figure.

8. Figure 8. Y axis uses “FREQ(%)” but “FREQ” is used elsewhere. Note the use of “/” in manuscript text but hyphen in X axis and figure caption (line 207). See also lines 204, 207 and replace hyphen with “/”. This figure also requires a legend for the frequency classes.

9. In the discussion are you able to provide an overview or summary of which groups you found to be more problematic than others and, for example, are likely in need for more comprehensive or updated identification guides.

6. PLOS authors have the option to publish the peer review history of their article (what does this mean?). If published, this will include your full peer review and any attached files.

Reviewer #1: No

---

## [Author Response · Author response to Decision Letter 0]

13 Oct 2020

Stribling and Leppo, 10/09/20

“Relationship of taxonomic error to frequency of observation” PONE-D-20-27394

(RESPONSES TO COMMENTS ARE IN ALL CAPS OR QUOTATION MARKS)

1. Lines 221, 226, 229, 230, 233-4, 239, 242 can you provide context to the taxonomic groups (e.g., order) the mentioned taxa belong, as done for Nais (line 222)? For a non-taxonomist or ecologist without knowledge of North America macroinvertebrates this would help immensely rather than the having to cross reference with supplementary files. I APPRECIATE THIS COMMENT. HOWEVER, IS IT NOT SUFFICIENT THAT THE TAXA MENTIONED IN THIS SECTION, AS REPRESENTATIVE AND SELECTED EXAMPLES, ARE GIVEN WITH THEIR CLASSIFICATION HIERARCY IN TABLE 4? I WOULD CONTEND THAT IT IS SUFFICIENT, AND ACTUALLY MAKES THE TEXT AN EASIER READ WITHOUT THE INLINE PARENTHETICALS. FOR CONSISTENCY, I DELETED THE HIERARCHY PARENTHETICAL FOR NAIS.

2. Although there is no mention to macroinvertebrate size (in relation to instar size) can you comment if instar size correlates to an increase in RPD error? THE STRIBLING ET AL. 2008 PAPER CITED GOES INTO SOME OF THE CAUSES AND MAGNITUDES OF ERROR. BUT YES, BODY SIZE, LIFE STAGE, AND SPECIMEN CONDITION ALL PLAY AN OUTSIZED ROLE IN IDENTIFICATION ERROR. THE PART OF THAT WHICH IS INTERESTING IS THAT TAXONOMISTS THAT ARE MORE EXPERIENCED AND WITH MORE IN-DEPTH TRAINING ARE ABLE TO OBTAIN SOLID RESULTS (=ACCURATE, CONSISTENT) MORE EASILY THAN THOSE WHO ARE LESS EXPERIENCED. I ADDED THE FOLLOWING TEXT IN THE DISCUSSION SECTION: “Another potential use of these results would be in helping target individual taxa for determining causes, beyond lack of familiarity, of higher error rates. A common cause is known to be specimens in poor condition and/or small body size (early life stages, or instars). An outcome of such an investigation might be to specify standard procedures for some taxa, including for sampling, handling, preservation, and identification. An example of this might be a requirement that all larval Chironomidae be slide-mounted for examination under a compound microscope. We do not necessarily advocate this, as slide-mounting is not consistently needed by all laboratories or taxonomists. Rather, we stress that the taxonomist use whatever method is needed to attain the target taxonomic level as defined by program or study goals. The goal in this case is not to require that all taxonomists (or taxonomic technicians) slide-mount all chironomid midges; rather, the goal is to acquire genus level data for the taxon. In some cases, slide-mounting might be needed, in others, it would not. Thus, the need for such actions would be determined on a case-by-case, taxon-by-taxon, or even taxonomist-by-taxonomist basis, but the goal of genus level data remains the same.”

3. Lines 222, 224, 232. Consistency required for expressing percentages. I AM UNSURE OF THE ISSUE HERE, AND WHAT CHANGES ARE BEING REQUESTED.

4. Figure 1. Can this axis extend to 100% as seen in Figure 5? DONE Do this also for Fig. 3. DONE The titles of X and Y axes are in upper case and not consistent with the other figures. CORRECTED

5. Figures 3, 4 and 5. There is considerably overlap in these three figures. Figure 3 is not necessary as this information is repeated in Figure 5. Although there is subtle difference in how these data points are plotted, they are essentially the same graph. Instead, can figures 4 and 5 be combined? For example, can each of the data points that correspond to a UFC categories be colour coded or superimposed on Figure 4? I BELIEVE ALL THREE FIGURES ARE NECESSARY FOR THE PAPER. FIGURE 3 PROVIDES THE READER WITH AN UNCOMPLICATED VISUALIZATION OF THE DATA DISTRIBUTION, FIGURE 4 IS A DEMONSTRATION OF THE CATEGORIES, AND FIGURE 5 ACCOMPLISHES THE SUGGESTED SUPERIMPOSITION. IT JUST SEEMS LIKE THERE IS BETTER CLARITY IN HOW THE UFC STRUCTURE IS PRESENTED IF ALL THREE ARE RETAINED. AS THAT IS THE CORE FOUNDATION OF THIS ANALYSIS, I BELIEVE IT IS BEST PRESENTED AS IT IS.

6. Figure 6. A 2D rather than a 3D plot would be preferred. A legend is required (1 = High confidence, common; 2 = High confidence, moderately common etc) to avoid the reader having to reference back to Figure 4. CORRECTION MADE; LEGEND ADDED TO CAPTION.

7. Figure 7. This figure needs to be produced in a similar format to Figure 1, i.e., consistent formatting of Y and X axis titles. As noted above, it is suggested a legend is added to this figure. I WILL ADD THE LEGEND TO THE CAPTION. HOWEVER, I THINK I DISAGREE WITH THIS SUGGESTION ABOUT THE BARCHART STYLE. THE TWO CHARTS ARE SHOWING DIFFERERNT KINDS OF DATA, AND BY USING STACKED BAR CHARTS/PERCENTAGES, MUCH SPACE IS SAVED. THAT IS, I FEEL CONFIDENT THIS IS THE MOST EFFICIENT TECHNIQUE FOR SHOWING HOW EACH OF THE MAJOR TAXA SORT AMONG THE CATEGORIES. 

8. Figure 8. Y axis uses “FREQ(%)” but “FREQ” is used elsewhere. THESE FIGURE PANELS ARE UNIQUE, AND I BELIEVE THEY WARRANT SOMEWHAT UNIQUE AXIS TITLES. I WOULD LIKE TO OPT TOWARD SPELLING THEM OUT, SUCH AS, FROM TOP TO BOTTOM: “Frequency of observation (FREQ)” and “Relative percent difference (RPD)”. Note the use of “/” in manuscript text but hyphen in X axis and figure caption (line 207). See also lines 204, 207 and replace hyphen with “/”. DONE. This figure also requires a legend for the frequency classes. I ADDED A LEGEND TO THE CAPTION FOR THIS, AS WITH THE PRIOR FIGURES.

9. In the discussion are you able to provide an overview or summary of which groups you found to be more problematic than others and, for example, are likely in need for more comprehensive or updated identification guides. THE FOLLOWING TEXT WAS ADDED TO THE DISCUSSION SECTION: “Higher level macroinvertebrate taxa in this analysis shown to have greater identification confidence and consistency are midges (Insecta: Diptera: Chironomidae), caddisflies (Insecta: Trichoptera), beetles (Insecta: Coleoptera), snails (Mollusca: Gastropoda), and stoneflies (Insecta: Plecoptera) (Fig 7), as they are mostly made up of finer level taxa with UFC1-3. Conversely, higher level taxa for which identification data seem to be more problematic (i.e., greater uncertainty) are bivalves (Mollusca: Pelecypoda) and Crustacea (Arthropoda). Each of these groups have a higher percentage of taxa in UFC4-5.”

---

## [Editor Report · Decision Letter 1]

23 Oct 2020

Relationship of taxonomic error to frequency of observation

PONE-D-20-27394R1

Dear Dr. Stribling,

We’re pleased to inform you that your manuscript has been judged scientifically suitable for publication and will be formally accepted for publication once it meets all outstanding technical requirements.

Kind regards,

Judi Hewitt

Academic Editor

PLOS ONE
---

## [Editor Report · Acceptance letter]

29 Oct 2020

PONE-D-20-27394R1 

Relationship of taxonomic error to frequency of observation 

Dear Dr. Stribling:

I'm pleased to inform you that your manuscript has been deemed suitable for publication in PLOS ONE. Congratulations! Your manuscript is now with our production department. 

Kind regards, 

on behalf of

Dr. Judi Hewitt 

Academic Editor

PLOS ONE